# Cross-Attention-Based Reflection-Aware 6D Pose Estimation Network for Non-Lambertian Objects from RGB Images

**Chenrui Wu** \*[ID]**, Long Chen and Shiqing Wu**

College of Mechanical Engineering, University of Shanghai for Science and Technology, Shanghai 200093, China
\* Correspondence: wuchenrui@usst.edu.cn

**Abstract:** Six-dimensional pose estimation for non-Lambertian objects, such as metal parts, is essential in intelligent manufacturing. Current methods pay much less attention to the influence of the surface reflection problem in 6D pose estimation. In this paper, we propose a cross-attention-based reflection-aware 6D pose estimation network (CAR6D) for solving the surface reflection problem in 6D pose estimation. We use a pseudo-Siamese network structure to extract features from both an RGB image and a 3D model. The cross-attention layers are designed as a bi-directional filter for each of the inputs (the RGB image and 3D model) to focus on calculating the correspondences of the objects. The network is trained to segment the reflection area from the object area. Training images with ground-truth labels of the reflection area are generated with a physical-based rendering method. The experimental results on a 6D dataset of metal parts demonstrate the superiority of CAR6D in comparison with other state-of-the-art models.

**Keywords:** 6D pose estimation; non-Lambertian objects; physical-based rendering; dense matching; PnP

## 1. Introduction

Six-dimensional pose estimation of objects is a fundamental problem in computer vision. The application scenarios of 6D pose estimation include automatic driving [1], robotic surgery [2], AR/VR [3], bin picking [4–6], and auto-assembly [7,8].

The 6D pose estimation problem can be fundamentally solved in two steps. The first step is finding correspondences between the objects in an image and objects in 3D models. The second step is the calculation of the 3D translation and 3D rotation with the Perspective-n-Points (PnP) algorithm [9] via the correspondences. Different manually designed key-point descriptors, such as SIFT [10], SURF [11], and ORB [12], are used to find the correspondences. However, these descriptors tend to find key points whose color gradients vary sharply in the image. When the objects are textureless, these descriptors cannot find reliable key points for computing the poses. To tackle this issue, some template-based methods [13,14] were proposed. These kinds of methods build templates of objects from different angles and distances. Several functions that can determine which template is the most similar to an object in an image have been designed to find the most suitable poses for the objects. These methods work well on textureless objects when they have intact appearances in the image, but it is easy for them to fail when the objects are occluded. Varying light conditions also have a great influence on pose estimation due to the changing color gradients of objects.

With the rapid development of deep learning technologies in recent years, convolutional neural networks (CNNs) have been used to solve 6D pose estimation problems, and they have achieved great improvements in comparison with conventional methods [15–17].

The direct way for CNNs to predict a pose is to first detect the 2D position of an object in an image, which can be done with object detection networks such as YOLO, SSD, and faster-RCNN. The features in the bounding box, which contains the information of the objects, are fed into a CNN structure to regress the six parameters of the pose. Similarly to

template-based methods, direct regression through CNNs cannot handle circumstances of occlusion. Occluded states vary too much for CNNs to learn to distinguish features.

To figure out this problem, semantic segmentation has been leveraged to first find all of the pixels that belong to the objects, and then a neural network is trained to vote for the 3D translation of the object. The 3D rotation expressed in the quaternion is then regressed through tiny sub-branch multi-layer perceptions (MLPs).

However, direct pose regression that uses sparse constraint signals has difficulties in leading neural networks with large numbers of parameters to efficiently predict the circumstances. In order to train CNNs with dense constraint signals, two-stage pipelines have been proposed in order to first extract key points from the image and then calculate the poses with the PnP method. This new framework can be seen as replacing conventional manually designed key-point descriptors with descriptors that are obtained via data-driven learning. CNNs can better describe the geometric features of objects than conventional human-made descriptors can. The performance of learned descriptors is acceptable even under occluded or varying light conditions, and state-of-the-art performance has been achieved.

Despite the success of dense matching methods in the 6D pose estimation of daily-life objects with Lambertian surfaces, current methods can easily fail when considering non-Lambertian objects, such as metal parts in an industrial environment. The refection of the non-Lambertian surface causes the appearance of such objects to change all the time, which heavily disturbs the prediction of the dense matching of neural networks.

In this paper, we propose a solution for the 6D pose estimation of non-Lambertian objects based on the detection of the reflection area. We build our model on top of a dense matching framework with an additional reflection-aware sub-module. We used physical-based rendering (PBR) to generate synthetic images with different levels of light strength, roughness, and coating (strength of the surface reflection). These images were used to train the reflection-aware sub-module to detect the reflection area of the object, which was removed during the prediction of the dense matching of the correspondences. A bi-directional cross-attention module is proposed to further improve the reflection segmentation and the dense matching accuracy of our proposed framework. The main contributions of this paper are summarized as follows:

(1) We propose a novel framework for the 6D pose estimation of objects with non-Lambertian surfaces. The framework leverages a reflection-aware module to prevent the dense matching of the correspondences from encountering disturbances from specular surfaces caused by light reflection.

(2) We use a simplified PBR model to synthesize virtual images for training the reflection-aware module. The synthetic images are automatically generated; this can save a huge amount of work in taking and labeling real images for training.

(3) We introduce a bi-directional cross-attention module into our framework to further improve the accuracy of the reflection segmentation and the dense matching.

(4) We demonstrate that our method outperforms other state-of-the-art methods on a 6D pose estimation dataset of metal parts.

## 2. Related Work

In this section, we briefly go through the 6D pose estimation methods that are based on deep learning.

### 2.1. Holistic Methods

This kind of method aims to estimate the 3D position and 3D rotation of an object in a single shot. PoseNet [15] uses a CNN architecture to directly regress a 6D camera pose from an RGB image. However, directly localizing the object's 3D translation is difficult due to the lack of depth information on the object. PoseCNN [16] first segmented objects from images; then, the segmented features were fed into a sub-branch to regress the center location and 3D rotation of each object. Direct regression of the 3D location is difficult due

to the nonlinearity of the rotation space. The authors of [18] discretized the rotation space into a classification task to solve this problem.

### 2.2. Key-Point Regression Methods

Unlike holistic methods, key-point regression methods adopt a two-stage pipeline. The pre-defined key points are first voted on by neural networks; then, the 6D pose of the object is computed with the PnP algorithm. PVNet [17] selected eight key points from an object's surface via the farthest-point-sampling algorithm. A voting-based key point scheme was used to determine the final locations of the key points. HybridPose [19] extended the idea of PVNet with more geometric constraints, such as the distance between the key points and the symmetry of the key points within symmetric objects. PVN3D [20] further applied the key-point regression idea to a 3D point cloud and showed its robustness to occlusions. CDPN [21] treated rotation and translation differently for their discrepant data distributions. The 3D translation was regressed based on scale-invariant translation, while the 3D rotation was acquired through a confidence map and the PnP method.

### 2.3. Dense Key-Point Matching Methods

Noticing that regressing key points has limitations when the key points are unseen in an image, some researchers tried to match the seen pixels with the 3D model. DPOD [22] mapped a 3D model into UV space as a 2D representation; then, the dense correspondences could be determined with a dense regression method. PSGMN [23] used a pseudo-Siamese graph-matching framework to directly match the pixels from an image to a 3D model. SurfEmb [24] learned dense, continuous 2D–3D correspondence distributions over the surfaces of objects from data with no prior knowledge of visual ambiguities, such as symmetry. Though the dense matching scheme achieved state-of-the-art performance on several datasets, it needs a predefined coarse pose and iteratively computes the final pose, and it cannot be used in real-time applications.

## 3. Proposed Approach

The task of 6D pose estimation is to compute the relative 3D translation and 3D rotation between the camera coordinate system and the object coordinate system taken from an RGB image given as an input. As we adopt the two-stage pipeline for pose estimation, we first find the correspondences between the image and the 3D model, and then calculate the pose through the image projection model with the PnP method from a multi-view geometry [25]:

$$\mathbf{p} = K[\mathbf{R} \mid \mathbf{t}]P, \tag{1}$$

with

$$K = \begin{bmatrix} f_x & 0 & c_x \\ 0 & f_y & c_y \\ 0 & 0 & 1 \end{bmatrix}, \tag{2}$$

where $\mathbf{p} = [u\ v\ 1]^\top$ is the 2D projection in the image of a 3D point $P = [x\ y\ z\ 1]^\top$ on the 3D model of the object. $K$ symbolizes the intrinsic parameters of the camera that provide the perspective transformation from the camera coordinate system to the image coordinate system. $\mathbf{R}$ and $\mathbf{t}$ are the 3D rotation and 3D translation, respectively, of the object coordinate *with respect to* the camera coordinate.

### 3.1. Network Architecture

Similarly to most recent research [21,22,24], we separate the pipeline into an object detection stage and a pose estimation stage, as shown in Figure 1. The input image is first fed into an object detector to detect objects, which are picked out through bounding boxes. Then, the zoom-in strategy [21] is used to fit the bounding box into a predefined size. In the pose estimation stage, we apply a U-Net-like structure as the backbone to extract pixel-

wise deep features $x^I \in \mathbb{R}^{\frac{H}{4} \times \frac{W}{4} \times d}$ from the cropped image. $H$ and $W$ denote the height and width of the cropped images, respectively. $d$ is the dimension of the deep features. The self-attention layers, on the other hand, extract node-wise deep features $x^M \in \mathbb{R}^{V \times d}$, with $V$ being the number of nodes from the 3D model of the object. $x^I$ and $x^M$ are then fed into the bi-directional cross-attention layers to fuse the features. The corresponding fused features are denoted as $f^I$ and $f^M$, respectively. $f^I$ contains pixels that do not belong to the object and pixels under reflection area. Two sub-branches with convolutional layers are designed to seperately predict the object segmentation $\mathcal{M}_o$ and the reflection segmentation $\mathcal{M}_r$. The final segmentation of the object $\mathcal{M}_f$ is then obtained through

$$\mathcal{M}_f = \mathcal{M}_o - (\mathcal{M}_o \cap \mathcal{M}_r) \tag{3}$$

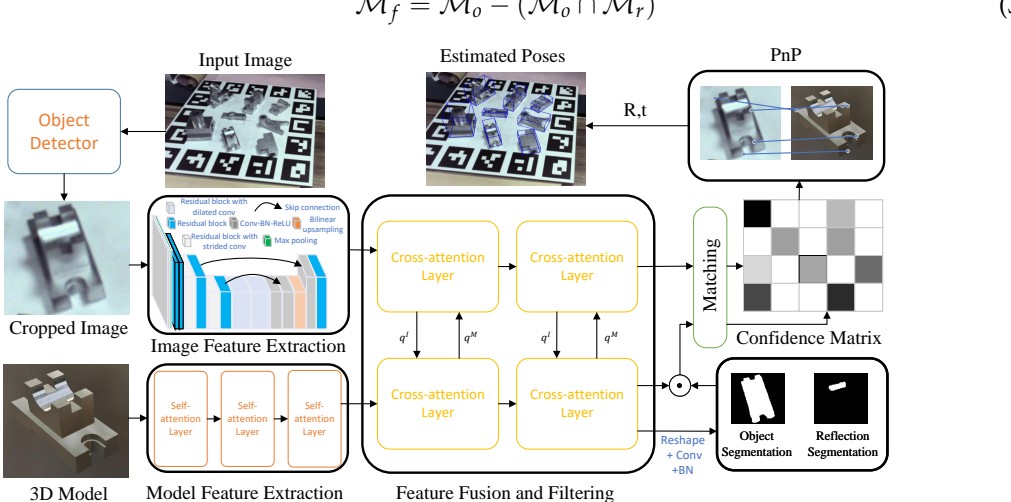

**Figure 1.** The architecture of the cross-attention reflection-aware network.

Only the fused image features $f^I$ that lie in the final segmentation mask are selected to calculate the scoring matrix with $f^M$:

$$\mathcal{S}(i,j) = \frac{1}{\sqrt{d}} \left\langle f_i^I, f_j^M \right\rangle \tag{4}$$

The softmax operator is then applied on both dimensions of $\mathcal{S}$ to convert the scoring matrix into a confidence matrix $\mathcal{C}$ [26].

$$\mathcal{C}(i,j) = Softmax(\mathcal{S}(i,\cdot)) \cdot Softmax(\mathcal{S}(\cdot,j)) \tag{5}$$

The confidence matrix decides the final correspondences of the feature matching. The 6D pose of the object is then calculated with the PnP method by using the correspondences.

### 3.2. Bi-Directional Cross-Attention Layers

Given the great improvements that transformer-based models have achieved in computer vision, we tried to look for a better way to combine the advantages of CNNs and transformers in the 6D pose estimation task. As a 3D model consists of vertices and edges, it is naturally suitable for the structure of the transformer. We use two conventional self-attention layers to extract deep features from the model. The self-attention operation is defined as:

$$\mathbf{q}_i^M = W_{\mathbf{q}^M} x_i^M \quad \mathbf{k}_j^M = W_{\mathbf{k}^M} x_j^M \quad \mathbf{v}_j^M = W_{\mathbf{v}^M} x_j^M \tag{6}$$

where $W_{\mathbf{q}^M}$, $W_{\mathbf{k}^M}$, and $W_{\mathbf{v}^M} \in \mathbb{R}^{d \times d}$ are learnable projection matrices. The superscript $M$ denotes that the parameters are from the 3D model. The query vector $\mathbf{q}^M$ and the key vector $\mathbf{k}^M$ are gathered to calculate the similarity of each feature through the dot product.

The similarity score weights the value vector $\mathbf{v}^M$ to be passed on. The input feature $x$ is then updated through

$$x^M \leftarrow x^M + \mathbf{MLP}(\sum_j a_{ij}^M \mathbf{v}_j^M) \tag{7}$$

where $a_{ij}^M = softmax(\mathbf{q}_i^M \mathbf{k}_j^{MT} / \sqrt{d})$ is the attention weight, and $\mathbf{MLP}(\cdot)$ is a two-layer fully connected network.

As shown in Figure 2, the bi-directional cross-attention module has a similar structure to that of a self-attention module. The difference mainly lies in that the input vectors $\mathbf{q}$ and $(\mathbf{k}, \mathbf{v})$ come from different inputs. Taking the cross-attention of the cropped image feature $x^I$ as an example, the output feature is defined as

$$x^I \leftarrow x^I + \mathbf{MLP}(\sum_j a_{ij}^I \mathbf{v}_j^I) \tag{8}$$

where $a_{ij}^I = softmax(\mathbf{q}_i^M \mathbf{k}_j^{IT} / \sqrt{d})$. The query vector and the key vector come from different inputs to generate mixed features for the mask filtering and the correspondence matching. Intuitively, it can be seen that each of the inputs (image and the 3D model) acts as a geometric filter for the other. The cross-attention structure forces the image to pay more attention to the features that are more related to the 3D model and vice versa. We also use the multi-head version of the attention layer, as is done in a conventional transformer [27].

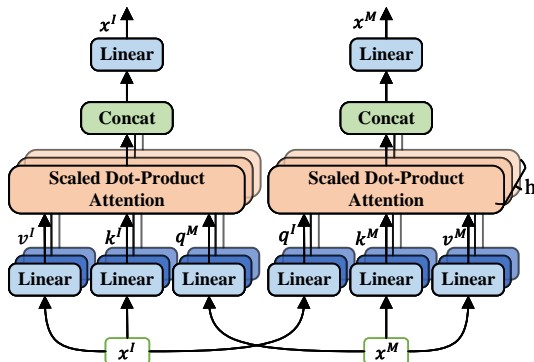

**Figure 2.** The structure of the cross-attention layer.

### 3.3. Reflection Label Acquisition

Currently, obtaining the reflection area of an object from a real image is very challenging. The inverse rendering problem [28] is ill conditioned due to the low rank of the input image. With the development of physical-based rendering technology, synthetic images have become very similar to real images. Many 6D pose estimation methods have shown that the estimation accuracy can be improved by a large margin with the help of PBR images. Thus, instead of obtaining the reflection area for training with inverse rendering methods, we use the PBR method to generate synthetic images with accurate reflection area labels.

Formally, as shown in Figure 3, assuming that the non-Lambertian object does not emit light, the outgoing radiance $L_o$ in direction $v$ from a surface point $p$ can be described by the following rendering equation:

$$L_o(p, v) = \int_\Omega f_r(p, v, \omega) L_i(p, \omega)(\omega \cdot n) \, d\omega \tag{9}$$

where $\Omega$ is the unit hemisphere centered at surface point $p$, with the $z$ axis being parallel to the surface normal $n$. $\omega$ represents the negative direction of incoming light. $f_r(p; v; \omega)$ represents the bi-directional reflectance function (BRDF) that relates to the material properties (e.g., color, roughness, and reflection) of surface $S$ at $p$, and $L_i(p; \omega)$ is the radiance coming toward $p$ from $\omega$. This integral computes the total effect of the reflection of every possible light ray that hits $p$ and bounces in direction $v$.

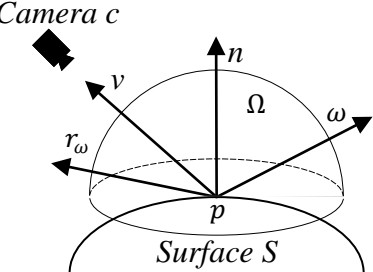

**Figure 3.** Notation and convention for viewpoint and illumination parameterization.

Our goal in using synthetic images is to mark out the reflection area of the object for network training. The PBR model is superior in the authenticity of its rendering, but $f_r$ combines diffuse light and specular light in one function, which makes it very difficult to figure out the reflection area. Instead of allowing an arbitrary BRDF $f_r$, the Phong reflection model [29] decomposes the outgoing radiance from point $p$ in direction $v$ into the diffusion and speculation functions. The view-independent part of the illumination is modeled by the following diffuse function:

$$I_{diffuse}(p) = \sum_{\omega \in \Omega} (\omega \cdot n) I_\omega \tag{10}$$

while the view-dependent part is modeled through the following speculation function:

$$I_{specular}(p, v) = \sum_{\omega \in \Omega} (r_{\omega,n} \cdot v)^\alpha I_\omega \tag{11}$$

According to the speculation function, we can directly mask out the reflection area. Note that it is important to find the reflection area for 6D pose estimation. As shown in Figure 4, the object segmentation is responsible for the whole structure of the object, while the reflection segmentation only focuses on partial high-light areas. As for dense matching, the pixels in the reflection area are supervised to match their correspondences to the 3D model. However, the image features of these pixels have bare information for the matching, which leads to the failure of finding correspondences at the inference stage and a bigger loss at the training stage.

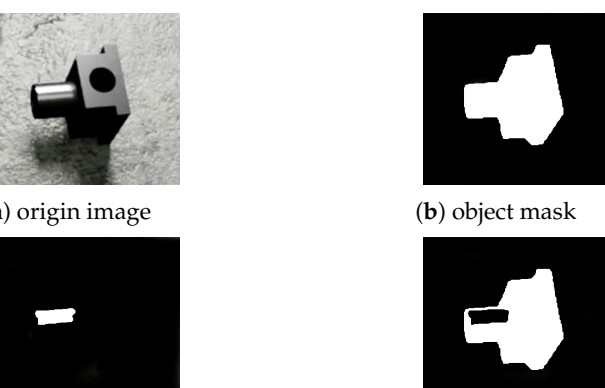

(**a**) origin image

(**b**) object mask

(**c**) reflection area

(**d**) mask without reflection

**Figure 4.** Illustration of the necessity of reflection removal. (**a**) The original rendered image. (**b**) The segmentation mask of the object. (**c**) The reflection area. (**d**) The object mask without the reflection.

## 4. Experiments

In this section, we first show the implementation details of our proposed network. Then, the MP6D dataset [30] and evaluation metrics are introduced. Qualitative and quantitive comparison results are given based on the MP6D dataset. Ablation studies are performed to show the effectiveness of the bi-directional cross-attention layers and the reflection-aware strategy of the proposed framework.

### 4.1. Implementation Details

Mesh model simplification: To reduce the memory usage of the network, we simplified the 3D mesh model of the objects from more than 5000 vertices to 2000 vertices through quadric edge collapse decimation in MeshLab [31]. This simplification helped the network train with a batch size of 16 for each GPU card.

Details of the network structure: We implemented our proposed network using Pytorch [32]. The image feature extraction module used Resnet18 from the Pytorch model-zoo as the backbone, which downsampled the input image at a scale of $8x$. An upsampling layer was applied to the module, which resulted in an output feature map with a size of $\frac{H}{4} \times \frac{W}{4}$. The height and width of the cropped image were both set to 240 pixels. The feature dimension $d$ was set to 64. The head count $h$ of the attention layer was set to 4 to reduce the memory cost.

Training strategy: As we only had the reflection segmentation labels from the synthetic images generated through PBR, we first trained our network using only the synthetic images for 20 epochs. After 20 epochs of training, the network could coarsely segment the reflection area. Then, we use real images from the MP6D dataset to train our network for another 20 epochs. Finally, another 10 epochs were processed to further refine the performance of the reflection segmentation task. We set the initial learning rate to 0.001 with a weight decay of 0.8 for every 10 epochs. For each object, we trained an independent model to achieve better performance. All models were trained using the Adam optimizer with 4 Nvidia RTX 2080Ti GPUs.

### 4.2. Dataset

In total, the MP6D dataset contained 77 video segments (20,100 frames in total) with occlusion and illumination changes. Twenty metal parts (15 parts from aluminum oxide material and 5 parts from copper oxide surface material) were collected from the industrial environment. The metal parts had sizes that ranged from 17 to 125 mm. All of the objects were textureless, symmetric, of a complex shape, and with a uniform color. The images were taken from different scenes to ensure the diversity of the data. The original dataset used PBR to render 50,000 synthetic images for training. However, these images did not contain the reflection segmentation labels. Therefore, we rendered synthetical images through PBR and calculated the reflection segmentation masks with the Phong model, as described in Section 3.3. The last 30% of the frames in each video segment were used as the test dataset to evaluate the performance of the 6D pose estimation methods.

### 4.3. Methods Used for Comparison and Metrics

We compared our method with other state-of-the-art 6D pose estimation methods, including PVNet [17] and PSGMN [23]. To quantitatively evaluate the performance, we used two commonly used metrics: average distance—symmetric (ADD-S) [16] and visible surface discrepancy (VSD) [33]. The ADD-S score was obtained with

$$ADD - S = \frac{1}{m} \sum_{x_1 \in \mathcal{M}} \min_{x_2 \in \mathcal{M}} \| (\mathbf{R}x_1 + \mathbf{t}) - (\tilde{\mathbf{R}}x_2 + \tilde{\mathbf{t}}) \| \tag{12}$$

where $m$ is the number of vertices in the 3D model. ADD-S finds the closest distance of a point to evaluate the performance in solving the symmetric situation, while the VSD metric pays more attention to the visible surface of the matching.

### 4.4. Comparison Results

Quantitative comparison: Table 1 gives the ADD-S scores and VSD scores of the different methods on the MP6D dataset. Apparently, our method achieved the best quantitative results for both metrics, demonstrating that our framework worked better with non-Lambertian objects. PVNet is a typical key-point regression method that regresses eight key points' 2D locations through a voting scheme. The main drawback of these kinds of methods is that they neglect the visibility of the key points. Therefore, the accuracy of the regressed key points goes down when they are invisible in the image. The method of [23] uses a dense matching strategy to find correspondences. The accuracy of this method is similar to that of our proposed method in reflection-free scenes, but it performs poorly when objects have reflective surfaces. Figure 5 shows the performance of the different methods in terms of the VSD score under varying light reflection conditions. It can be seen that our proposed method remained stable as the reflection area increased. The other methods suffered from a large decline with the increasing ratio of the reflection area.

**Table 1.** Quantitative evaluation of 6D poses in the MP6D dataset. The highest score for each object is shown in bold.

| | PVNet [17] | | PSGMN [23] | | Ours | |
|---|---|---|---|---|---|---|
| **Object** | **ADDS** | **VSD** | **ADDS** | **VSD** | **ADDS** | **VSD** |
| Obj_01 | **73.1** | 57.3 | 70.4 | 58.9 | 72.3 | **62.1** |
| Obj_02 | 65.9 | 58.9 | 74.2 | 57.6 | **81.5** | **60.3** |
| Obj_03 | 51.2 | 29.1 | 61.4 | 28.5 | **63.4** | **32.1** |
| Obj_04 | **60.1** | 48.5 | 53.2 | 51.7 | 53.3 | **56.2** |
| Obj_05 | 56.6 | 46.4 | 60.8 | 41.8 | **61.5** | **60.2** |
| Obj_06 | 62.2 | 46.2 | 68.7 | 61.8 | **71.2** | **63.7** |
| Obj_07 | **73.3** | 62.4 | 75.7 | 66.8 | 72.2 | **72.3** |
| Obj_08 | 70.3 | 54.4 | 70.1 | 51.2 | **74.4** | **55.8** |
| Obj_09 | 58.8 | **29.5** | 64.0 | 19.8 | **68.1** | 24.5 |
| Obj_10 | 61.1 | 45.7 | **70.6** | 61.7 | 53.4 | **65.3** |
| Obj_11 | 50.5 | 32.2 | 61.5 | 32.8 | **74.6** | **38.3** |
| Obj_12 | 34.5 | 29.5 | 69.3 | 41.2 | **70.3** | **43.5** |
| Obj_13 | 63.3 | 26.4 | 70.2 | 39.5 | **78.2** | **42.2** |
| Obj_14 | 58.9 | 39.8 | 70.1 | 40.9 | **73.9** | **44.1** |
| Obj_15 | 46.7 | 5.5 | 59.2 | 8.4 | **62.5** | **10.2** |
| Obj_16 | 57.2 | 25.3 | **63.5** | 27.5 | 59.8 | **31.8** |
| Obj_17 | 64.0 | 56.1 | 72.2 | 57.3 | **79.6** | **59.5** |
| Obj_18 | 64.9 | 59.2 | 51.7 | 68.4 | **80.5** | **80.2** |
| Obj_19 | 62.4 | 40.3 | 70.1 | 34.8 | **71.3** | **41.9** |
| Obj_20 | 59.3 | 50.1 | 69.4 | 53.8 | **73.4** | **42.8** |
| Average | 59.7 | 42.1 | 66.3 | 45.2 | **69.8** | **49.3** |

Qualitative comparison: Figure 6 visually compares the 6D pose estimation results of our proposed method with those of the other state-of-the art methods. Our method was able to successfully predict the pose under reflection, while the other methods suffered from the high-light areas that lacked rich features. We found that high-light areas may lead to incorrect correspondence matching. The cross-attention layers helped to filter out these reflection areas and infer feature correlations. The reflection segmentation branch further prevented the reflection areas from disturbing the 6D pose estimation process. We also show some examples of the pose estimation results from the test set of the MP6D dataset in Figure 7. This shows that our method works robustly under different challenging conditions, such as illumination variation, occlusion, and light reflection.

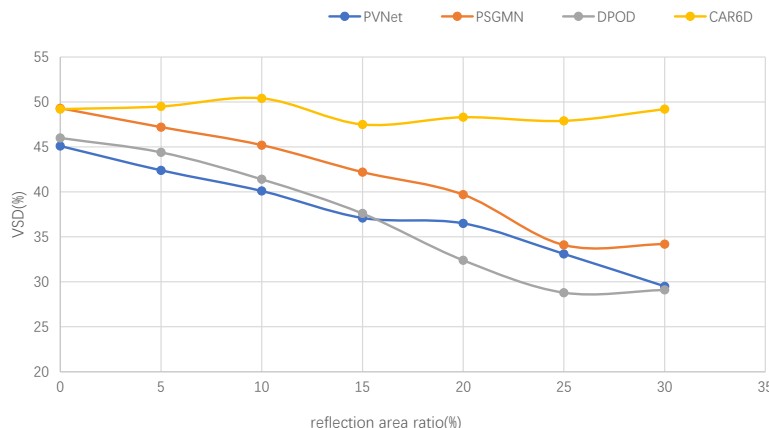

**Figure 5.** Comparison among PVNet, PSGMN, DPOD, and our proposed CAR6D method in terms of the VSD performance under different levels of light reflection.

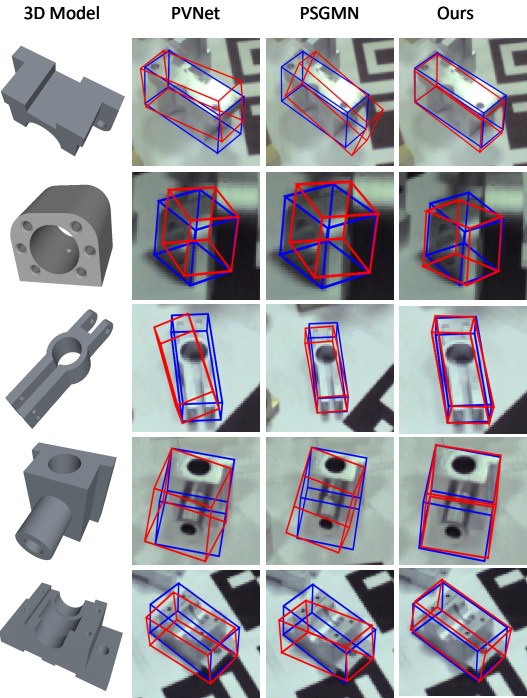

**Figure 6.** Comparison among PVNet, PSGMN, and our proposed CAR6D method. The ground-truth poses are presented in blue boxes, while the estimated poses are shown in red boxes.

*4.5. Ablation Studies*

We performed ablation studies on the model structure and the segmentation mask to verify the effectiveness of the cross-attention layers and the reflection segmentation.

We first replaced the cross-attention layers with self-attention layers. This action made the 3D model features and image features independent of each other. The result of this action is shown in Table 2. $mAP_*$ denotes a ratio that is used as follows: When the intersection over union (IOU) of the predicted segmentation mask to the ground-truth mask is higher than a given ratio (50% and 75% were used in the evaluation), the mask is regarded as valid. The cross-attention layers help the network segment the object and reflection, with an increase of more than 8% in both object segmentation and reflection segmentation in terms of $mAP_{0.75}$.

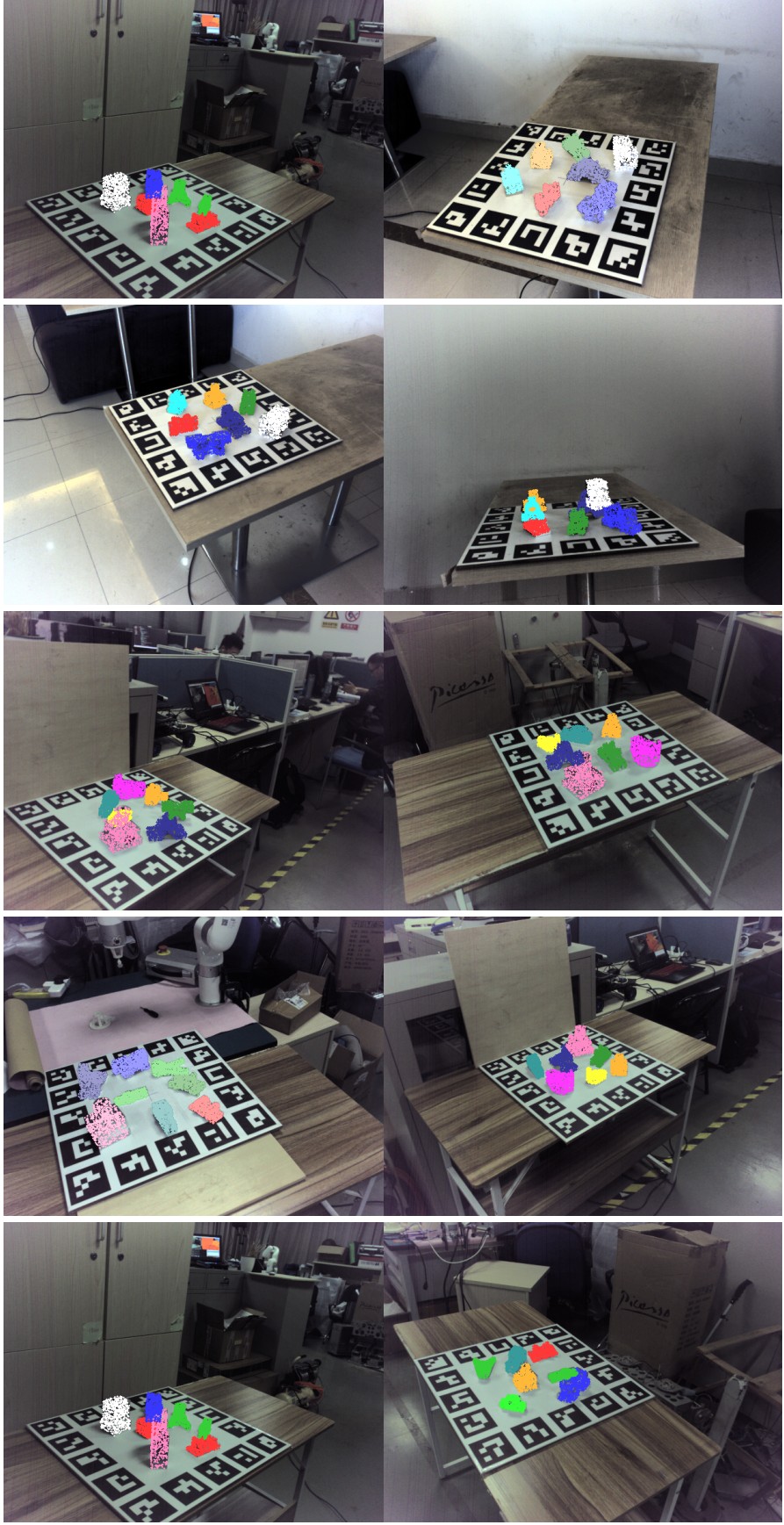

**Figure 7.** Some qualitative results of our proposed method.

**Table 2.** A comparison of the cross-attention layers and the self-attention layers.

| Model Structure | Object Segmentation | | Reflection Segmentation | |
|---|---|---|---|---|
| | $mAP_{0.5}$ | $mAP_{0.75}$ | $mAP_{0.5}$ | $mAP_{0.75}$ |
| Cross-attention layers | 83.13 | 80.62 | 72.68 | 69.52 |
| Self-attention layers | 80.79 | 72.58 | 67.77 | 60.21 |

We also tested the impact of reflection segmentation on non-Lambertian surfaces. As shown in Table 3, if only self-attention layers were used without reflection segmentation, the results were inferior to those of PSGMN [23], which used a deeper graph neural network for the feature extraction of the 3D model. With the refection segmentation, our model outperformed the other methods by 0.9% in terms of VSD. With both cross-attention and refection segmentation, our model achieved a 6.2% improvement, thus demonstrating the effectiveness of the proposed framework.

**Table 3.** Ablation study on the model structure and segmentation mask.

| Model | Self-attention layers | ✓ | ✓ | | |
|---|---|---|---|---|---|
| Structure | Cross-attention layers | | | ✓ | ✓ |
| Reflection | with | | ✓ | | ✓ |
| Segmentation | without | ✓ | | ✓ | |
| Results | ADD-S | 61.7 | 66.5 | 67.5 | 69.8 |
| | VSD | 43.1 | 46.1 | 44.6 | 49.3 |

## 5. Conclusions

In this paper, we proposed CAR6D, a cross-attention-based reflection-aware network for 6D pose estimation. We solved the problem of light reflection in pose estimation for non-Lambertian objects in two ways. We proposed a framework for training the network that is aware of the reflection area. PBR and the Phong model were used to generate synthetic images with reflection segmentation labels for training. We also proposed a bi-directional cross-attention structure to fully fuse the image features with the model features. The experimental results showed that the proposed method outperformed the other state-of-the-art methods on MP6D, a dataset of non-Lambertian metal parts. We note that the current way to segment the reflection area mainly depends on the data distribution. Further study on the elimination of high-light areas should be carried out in order to achieve better performance in 6D pose estimation for non-Lambertian objects.

**Author Contributions:** C.W.: conceptualization, methodology, writing—original draft, software, and validation. L.C.: formal analysis, investigation, and supervision. S.W.: writing—review and editing. All authors have read and agreed to the published version of the manuscript.

**Funding:** The authors gratefully acknowledge the financial support from the National Natural Science Foundation of China (Nos. 52105525, 52075340, and 52005338).

**Data Availability Statement:** Publicly available datasets were analyzed in this study. This data can be found here: https://github.com/yhan9848/MP6D (accessed on 19 September 2022).

**Conflicts of Interest:** The authors declare no conflict of interest.

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
