# Peer review of "Cross-Attention-Based Reflection-Aware 6D Pose Estimation Network for Non-Lambertian Objects from RGB Images"

_machines, doi:10.3390/machines10121107_

Round 1

Reviewer 1 Report

Pose estimation of non-Lambertian objects is a real-world problem. This manuscript presents a model for 6D pose estimation of such objects. The proposed model is built on top of the dense matching framework with an additional reflection-aware sub-module. Firstly, the authors generate synthetic images, using physical-based render (PBR), which is fed to the reflection-aware sub-module to detect the reflection area. To further improve the reflection segmentation and the dense matching accuracy authors use a bi-directional cross-attention module. Authors claim to achieve superior results compared to the previous approaches.

The manuscript is well-written and presented in a comprehensive manner. However, there are a few typos and sentence structure mistakes that need to be addressed, eg on line 84 "methods aims" should be corrected to "methods aim". My suggestion is to carefully proofread the manuscript before re-submission.

Author Response

Point 1: However, there are a few typos and sentence structure mistakes that need to be addressed, eg on line 84 "methods aims" should be corrected to "methods aim". My suggestion is to carefully proofread the manuscript before re-submission.

Response 1:  Thank you for the comment. According to the comment, we carefully proofread the manuscript and fix the typos and the sentence structure mistakes. The modifications are shown in red in the revised manuscript.

Reviewer 2 Report

This paper is well written and the method looks correct. I only have a few concerns:

In the section of implementation details, the authors claimed "[...] To reduce the memory usage of the network, we simplify the 3D mesh model of the object to 2000 vertices through quadric edge collapse decimation in MeshLab [...]", what is the raw resolution of the mesh? Why do you select 2000 vertices?

In Table 1, I cannot recognize which method performed the best. Necessary information should be given.

In Figure 5, what’s the meaning of the red or blue box? Prediction or ground truth?

Author Response

Point 1: In the section of implementation details, the authors claimed "[...] To reduce the memory usage of the network, we simplify the 3D mesh model of the object to 2000 vertices through quadric edge collapse decimation in MeshLab [...]", what is the raw resolution of the mesh? Why do you select 2000 vertices?

Response 1:  Thank you for the comment. The raw resolution of the mesh differs from 5000 to 8000 vertices for different objects in MP6D. We select 2000 vertices to save the GPU memory usage and speed up the training procedure. The number 2000 is set according to the memory size of the 2080ti GPU. We add this explanation in the section of implementation details.

Point 2: In Table 1, I cannot recognize which method performed the best. Necessary information should be given.

Response 2:  Thank you for the comment. According to the comment, we bold the number of the best score in Table 1 to show the best performance more clearly.

Point 3: In Figure 5, what’s the meaning of the red or blue box? Prediction or ground truth?

Response 3:  Thank you for the comment. According to the comment, we add the explanation of the red box and blue box in Figure 5.